

# Phylogenetic relationships and evolutionary history of the greater horseshoe bat, *Rhinolophus ferrumequinum*, in Northeast Asia

Tong Liu[1], Keping Sun[1], Yung Chul Park[2] and Jiang Feng[1]

[1] Jilin Provincial Key Laboratory of Animal Resource Conservation and Utilization, Northeast Normal University, Changchun, China
[2] Department of Forest Environment Protection, College of Forest and Environmental Science, Kangwon National University, Chuncheon, South Korea

## ABSTRACT

The greater horseshoe bat, *Rhinolophus ferrumequinum*, is an important model organism for studies on chiropteran phylogeographic patterns. Previous studies revealed the population history of *R. ferrumequinum* from Europe and most Asian regions, yet there continue to be arguments about their evolutionary process in Northeast Asia. In this study, we obtained mitochondrial DNA cyt *b* and D-loop data of *R. ferrumequinum* from Northeast China, South Korea and Japan to clarify their phylogenetic relationships and evolutionary process. Our results indicate a highly supported monophyletic group of Northeast Asian greater horseshoe bats, in which Japanese populations formed a single clade and clustered into the mixed branches of Northeast Chinese and South Korean populations. We infer that *R. ferrumequinum* in Northeast Asia originated in Northeast China and South Korea during a cold glacial period, while some ancestors likely arrived in Japan by flying or land bridge and subsequently adapted to the local environment. Consequently, during the warm Eemian interglaciation, the Korea Strait, between Japan and South Korea, became a geographical barrier to Japanese and inland populations, while the Changbai Mountains, between China and North Korea, did not play a significant role as a barrier between Northeast China and South Korea populations.

Corresponding authors
Keping Sun, sunkp129@nenu.edu.cn
Jiang Feng, fengj@nenu.edu.cn

## INTRODUCTION

During the past 2.5 million years, Earth has been in glacial and interglacial periods of the Quaternary Ice Age (*Capinera, 2011*). Climatic change and the existence of refugia have influenced effective population size and demographic history of organisms and left genetic signatures in current populations (*Avise, 2000*; *Hewitt, 2000*; *Qu et al., 2014*). The effective population size of organisms may decrease during the glacial period or remain stable or grow due to intermittent gene flow between refugia during warming periods (*Li et al., 2009*; *Qu et al., 2014*). The large volume of accumulated ice during the most recent Quaternary glaciation period caused a worldwide sea level drop by 120–140 m below the

present sea level (*Lambeck, Esat & Potter, 2002*). Land bridges appeared in several parts of the world, which inevitably led to range changes for most living organisms (*Hewitt, 2000*). Consequently, organisms adapted to different environments and new neighbors, causing genetic variation, both through selection and drift, and ultimately speciation (*Harrison, 1993*; *Hewitt, 2000*).

The greater horseshoe bat, *Rhinolophus ferrumequinum* (Rhinolophidae, *Rhinolophus*), is widely distributed in northern Africa, southern Europe, and Asia (*Csorba, Ujhelyi & Thomas, 2003*). In China, it ranges from northeastern to southwestern regions (*Wang, 2003*). Previous studies have revealed the impact of glaciations on their geographic patterns in Europe and most Asian regions (*Rossiter et al., 2000*; *Rossiter et al., 2007*; *Bilgin et al., 2009*; *Flanders et al., 2009*; *Flanders et al., 2011*), but little is known about this species from Northeast Asia. *Flanders et al. (2009)* and *Flanders et al. (2011)* showed that *R. ferrumequinum* from the Jilin Province of China was affiliated with those from Japan based on the mtDNA *ND2* gene, which suggests greater horseshoe bats might move between east China and Japan using South Korea as a stepping-stone (*Flanders et al., 2009*) or by the Korean Peninsula-Japanese land bridge during glacial periods (*Flanders et al., 2011*). However, no Korean samples and only one sample from Jilin Province were analyzed in the *Flanders et al. (2009)* and *Flanders et al. (2011)* studies. Their results indicated that the Jilin Province sample was located in the East clade of China, whereas *Sun et al. (2013)* showed that all samples from the Jilin Province were divided into another Northeast clade based on the mtDNA D-loop region. Therefore, it is necessary to combine more samples from Northeast China, South Korea and Japan to investigate the population evolutionary process of *R. ferrumequinum* in Northeast Asia.

Natural landscape features, such as mountains and rivers, can function as genetic boundaries and shape the population structure of animals by hindering dispersal and gene flow (*Funk et al., 2008*; *Bilgin et al., 2009*; *Fünfstück et al., 2014*). For *R. ferrumequinum* in Northeast Asia, the Yalu River and Changbai Mountains separated the populations from Northeast China and Korean Peninsula. Additionally, the Korea Strait separated the populations in Korea from those in Japan. *Koh et al. (2014)* considered that the Yalu River and Changbai Mountains did not play a role as physical barriers for Korean and adjacent Northeast Chinese populations in *R. ferrumequinum* based on mtDNA cyt *b* gene. However, only one sample from Northeast China was included in their analyses.

In this study, we collected and sequenced mtDNA cyt *b* and blank D-loop sequences of additional *R. ferrumequinum* samples from Northeast China and South Korea, and analyzed them with all of the previously published mtDNA sequences from China, Japan and South Korea. Our aims were to (i) clarify the phylogenetic relationships of *R. ferrumequinum* in Northeast Asia, (ii) infer the evolutionary process in Northeast Asia and (iii) detect whether the Changbai Mountains and Korea Strait act as geographical barriers for *R. ferrumequinum*.

## MATERIALS AND METHODS

*R. ferrumequinum* individuals were sampled from Northeast China and South Korea. Twenty-two and 49 individuals were used to sequence mtDNA cyt *b* and D-loop region,

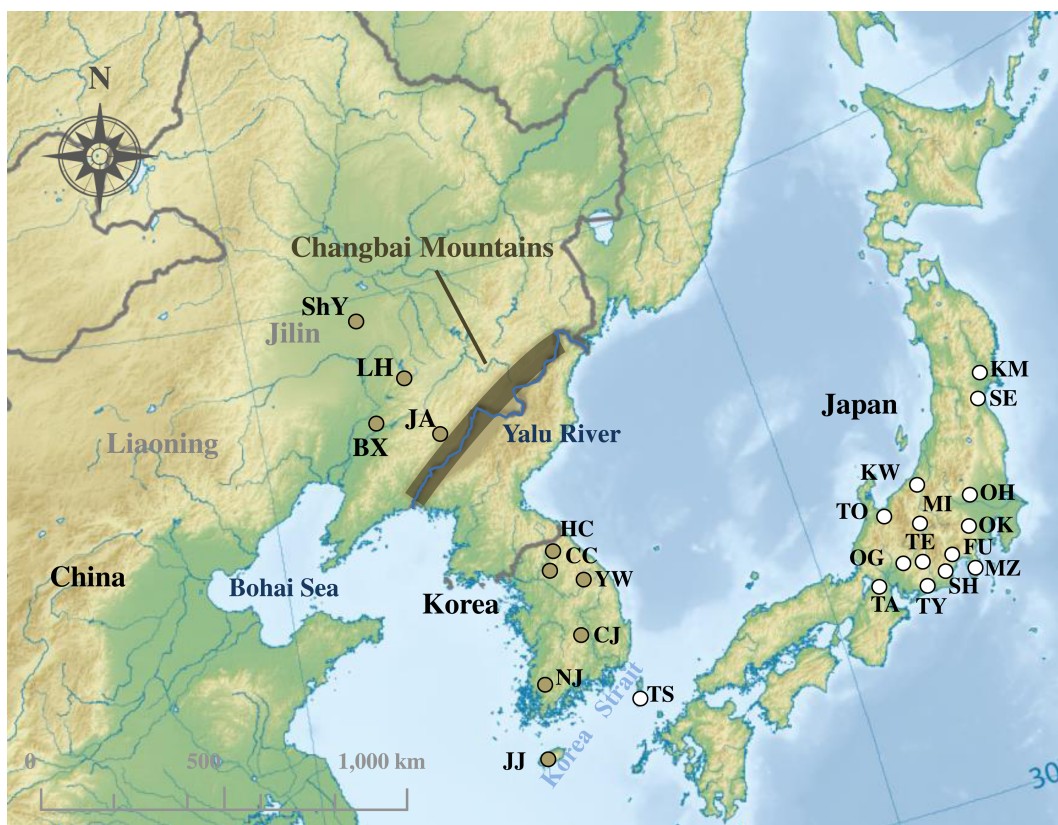

**Figure 1** **Sampling localities of *Rhinolophus ferrumequinum* in this study.** The colors of sampling points fit with clades identified in Fig. 2. Locality codes are identical to those in Table S1. Map uploaded to Wikimedia Commons by Ksio unde the GNU Free Documentation License.

respectively. A total of 76 sequences (63 cyt *b* sequences and 13 D-loop sequences) of *R. ferrumequinum* were collected from Japan, South Korea and China (Fig. 1; Table S1). For the D-loop region, our study did not include the sequences from Japan because no D-loop sequence of Japanese *R. ferrumequinum* was deposited in GenBank. All field studies were approved by National Animal Research Authority in Northeast Normal University, China (approval number: NENU-20080416).

Previous studies and this study show South Korean, Japanese and Northeast Chinese *R. ferrumequinum* have very low genetic divergence (*Sakai, Kikkawa & Tsuchiya, 2003*; *Koh et al., 2014*). Therefore, we regard bats from South Korea, Japan or Northeast China as a single geographic population.

## DNA extraction and amplification

Bat wing membrane tissues were taken and stored at 95% ethanol solution until genome extraction by the UNIQ-10 Column Animal Genomic DNA Isolation Kit (SK1205) (Sangon, China). Mitochondrial DNA cyt *b* and D-loop region were amplified by polymerase chain reaction (PCR) using universal primers L14724 and H15915 (*Irwin, Kocher & Wilson, 1991*) and P and E (*Wilkinson & Chapman, 1991*), respectively. Amplified products were purified and sequenced by Shanghai Sangon Biotechnology Co., Ltd. Sequences were edited

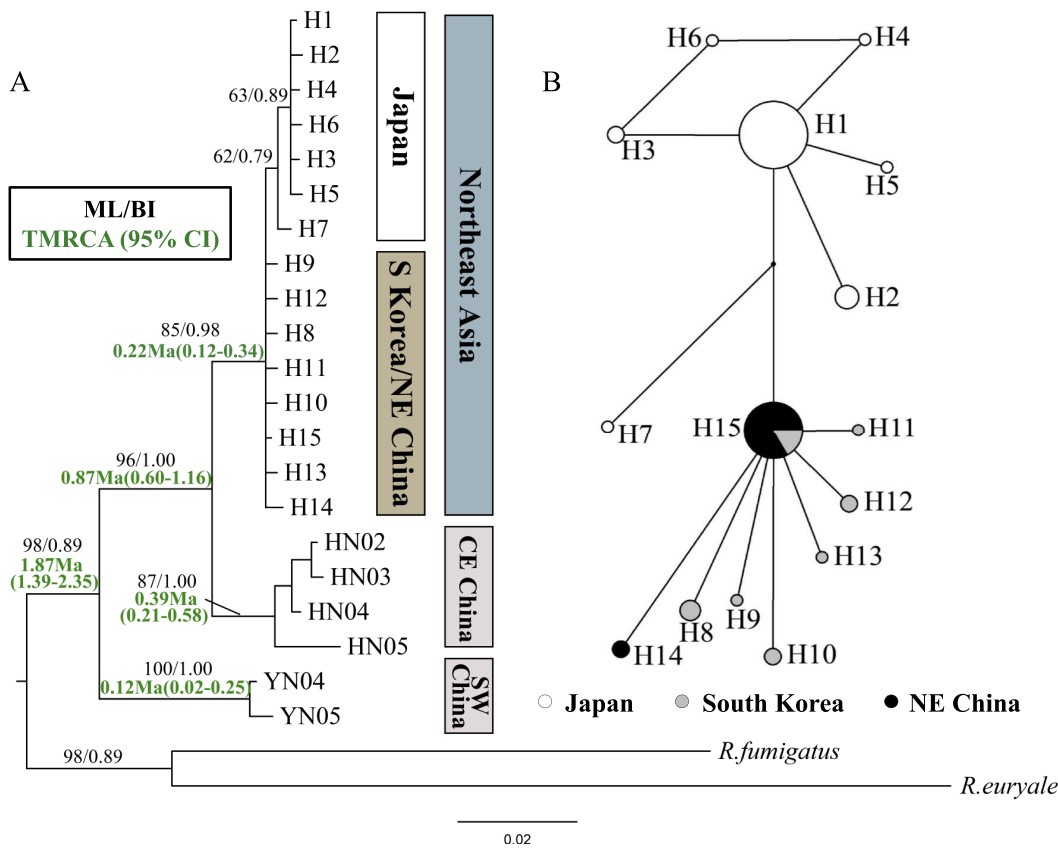

**Figure 2** **Phylogenetic trees and network for *Rhinolophus ferrumequinum* populations based on cyt *b* haplotypes.** (A) Phylogenetic trees constructed by ML and BI methods. (B) Median-joining network for the East Asian *R. ferrumequinum* haplotypes. The circle size is proportional to the frequency of that haplotype. Small black dots represent missing haplotypes. Locality codes and haplotype are described in Tables S1 and S2, respectively.

and aligned using Geneious v8.0.2 (*Kearse et al., 2012*), and then were assembled by eye. Sequence data were deposited in GenBank (accession number: KX237527–KX237538 and KX237546).

## Genetic diversity

Haplotype diversity, nucleotide diversity and polymorphic sites of each population were calculated based on cyt *b* and D-loop sequences, respectively. The gene flow was calculated using cyt *b* gene sequences based on the equation: Fst = $1/(1 + 4\,Nm)$. All calculations were carried out using DnaSP v4.0 (*Rozas et al., 2003*). Pairwise distances between populations were obtained using Kimura-2-Parameter (K2P) distance model (*Kimura, 1980*), with 1,000 bootstrap replications, using MEGA v5.0 (*Tamura et al., 2013*).

## Phylogenetic analysis

Phylogenetic trees of cyt *b* and D-loop were reconstructed using maximum likelihood (ML) methods in PhyML v3.1 (*Guindon et al., 2010*) and Bayesian Inference (BI) in MrBayes v3.2.2 (*Ronquist & Huelsenbeck, 2001*; *Ronquist & Huelsenbeck, 2003*). PhyML starts with

a BioNJ tree by default (*Gascuel, 1997*), and statistical support for branching patterns was estimated by bootstrap with 1,000 replicates. BI was run with four Markov Chains Monte Carlo (MCMC), each of $1 \times 10^7$ generations, sampled every 100 generations. Two congeneric species from the Afro-Palearctic clade, *Rhinolophus euryale* (GenBank nos. EU436671 and KF031268) and *R. fumigatus* (GenBank nos. EU436678 and KU531336) were used as outgroups.

ML and BI methods are sensitive to nucleotide substitution models, which can be estimated by jModelTest v0.1 (*Posada, 2008*). According to the Akaike information criterion (AIC) (*Posada & Buckley, 2004*), the HKY + G (transition/transversion = 12.9820; gamma shape = 0.1510) and HKY + G (transition/transversion = 4.5832; gamma shape = 0.1840) were selected for cyt *b* and D-loop, respectively.

The program NETWORK v4.6 (*Bandelt, Forster & Röhl, 1999*) was used to build a maximum parsimony network using the median-joining method which simplified same-possible trees and removed redundant nodes and connections (*Puizina et al., 2013*) as well as represented the intraspecific genetic variation (*Posada & Crandall, 2001*).

In order to estimate divergent time, the time to the most recent common ancestor (TMRCA) was estimated using BEAST v1.6 (*Drummond & Rambaut, 2007*) under a strict clock and a constant-size tree prior. The chain was run for $1 \times 10^7$ generations, with the ESS values >200 taken as evidence for convergence. Cyt *b* gene was chosen to calculate TMRCA because of its moderate evolutionary rate. A mean substitution rate of 1.3% per million years (*Nabholz, Glemin & Galtier, 2008*), used in *Hipposideros turpis* complex (*Thong et al., 2012*), *Hipposideros commersoni* (*Rakotoarivelo et al., 2015*) and *Myotis nattereri* complex (*Puechmaille et al., 2012*), was used in this analysis.

## Demographic analysis

The demographic expansion of Northeast Asian *R. ferrumequinum* was examined in Arlequin v3.1 (*Schneider, Roessli & Excoffier, 2000*). Tajima's *D* (*Tajima, 1989*) and Fu's *Fs* tests (*Fu, 1997*) were employed to confirm neutral expectation based on 1,000 coalescent simulations. Significant negative Tajima's *D* and Fu's *Fs* values indicate a sudden expansion, whereas significant positive values indicate processes such as population subdivision or recent bottlenecks. When the values are nearly zero, they represent a population of constant size (*Liao et al., 2010*) The raggedness index (Hri; *Harpending, 1994*) and sum of squared deviations (SSD; *Schneider & Excoffier, 1999*) were generated with 10,000 replicates parametric bootstrapping. Hri was calculated to describe the smoothness of observed mismatch distribution. The small value means a population has experienced a sudden expansion event, and the higher value indicates a stationary or bottlenecked population (*Harpending, 1994*; *Liao et al., 2010*). SSD value was used to describe the goodness-of-fit of observed mismatch distribution to that expected under the spatial expansion model. A non-significant SSD value ($P_{SSD} > 0.05$) suggests a good fitness (*Excoffier, Laval & Schneider, 2005*). The mismatch distribution graphs were drawn in DnaSP v4.0. A smooth or unimodal mismatch distribution indicates an expanded population, while a ragged or multimodal distribution indicates a more stable population (*Rogers & Harpending, 1992*; *Flanders et al., 2011*).

**Table 1  Genetic diversity of *Rhinolophus ferrumequinum* in Northeast Asia.**

|  | $N_s$ | $N_h$ | $N_{ss}$ | $h$ (cyt $b$/D-loop) | $\pi$ (cyt $b$/D-loop) |
|---|---|---|---|---|---|
| NE China | 22/43 | 2/7 | 2/6 | 0.173/0.564 | 0.030/0.226 |
| South Korea | 14/8 | 7/4 | 6/3 | 0.879/0.648 | 0.116/0.201 |
| Japan | 43/- | 7/- | 6/- | 0.408/- | 0.047/- |
| Northeast Asia | 79/51 | 15/10 | 15/9 | 0.735/0.573 | 0.143/0.229 |

**Notes.**

$N_s$, the number of sequences; $N_h$, the number of haplotypes; $N_{ss}$, the number of segregating sites; $h$, haplotype diversity; $\pi$, nucleotide diversity;  -,  missing data.

If the expansion was detected, the time of expansion in generations ($t$) can be estimated by the equation, $\tau = 2ut$, where $\tau$ (tau) is the time to expansion in mutational units and $u$ is the mutation rate per generation for the DNA sequence being studied. Cyt $b$ mutation rate is 1.3% per million years (*Nabholz, Glemin & Galtier, 2008*), and the generation time is two years (*Ransome, 1995*).

## RESULTS

### Genetic diversity and divergence

A total of 85 sequences based on the cyt $b$ gene (1,140 bp) and 62 sequences based on D-loop region (465 bp) were obtained and analyzed (Table S1). For the cyt $b$ gene, 15 different haplotypes were identified from 79 sequences of Northeast Asian *R. ferrumequinum*. The Japanese population had 7 unique haplotypes, while Chinese and South Korean populations shared haplotype H15, which was the most shared haplotype. For D-loop region, 10 unique haplotypes were identified from 51 sequences of *R. ferrumequinum* in Northeast Asia. H9 was shared by individuals from Ji'an and Benxi in Northeast China, whereas H10 was shared by most individuals and populations (including individuals in all four localities of China and some individuals in South Korea) (Table S2).

Within cyt $b$ and D-loop haplotypes, there were 15 (1.3%) and 9 (2.0%) polymorphic sites, respectively, and 10 (0.87%) and 5 (1.1%) parsimonious informative sites, respectively. Genetic diversity of the South Korean population was the highest, while that of the Northeast Chinese population was the lowest (Table 1).

The cyt $b$ divergence of Northeast Asian *R. ferrumequinum* was lower than 1%. The average K2P distances between populations from Northeast China and South Korea (0.07%) were lower than those between Japanese and other Northeast Asian populations (0.21–0.26%). Furthermore, the gene flow between South Korea and Northeast China (Nm $\geq$ 3) was highest, which was enough to prevent genetic divergence caused by genetic drift (*Slatkin, 1987*; *Yang, Ma & Wu, 2011*). However, the gene flow levels between Japanese and the other Northeast Asian populations were low (Table 2).

### Phylogenetic relationships and TMRCA

The ML and BI tree topologies based on cyt $b$ gene produced highly concordant phylogenetic relationships. All samples from Northeast Asia formed a highly supported monophyletic clade (Fig. 2). In the tree, the relationship between Northeast Chinese and South Korean

**Table 2** Average K2P distance (%) and gene flow of *Rhinolophus ferrumequinum* based on cyt *b* sequences.

| Population | NE China | South Korea | Japan | Henan | Yunnan |
|---|---|---|---|---|---|
| NE China | | 3.11 | 0.6 | 0.06 | 0.01 |
| South Korea | 0.7 | | 0.11 | 0.06 | 0.01 |
| Japan | 0.21 | 0.26 | | 0.05 | 0.01 |
| Henan | 1.44 | 2.01 | 2.15 | | 0.03 |
| Yunnan | 3.96 | 4.02 | 4.17 | 4.27 | |

**Notes.**

Nm: above the diagonal; Average K2P distance (%): below the diagonal.

haplotypes was much less resolved, likely due to only a few mutations. Japanese haplotypes formed their own clade, but with relatively low bootstrap values (62/79% in ML/BI, respectively) (Fig. 2A). The Japanese clade clustered into the mixed branches of South Korean and Northeast Chinese haplotypes. The haplotypes of Northeast Asia were sister to those of the Central-East China (Fig. 2A). The haplotype network showed similar relationships with the phylogenetic trees, while showing the relationship between the haplotypes more clearly (Fig. 2B).

The noncoding D-loop region generally provides sufficient variation for studies at intraspecific level (*Qu et al., 2009*). However, in the phylogenetic tree and haplotype network based on D-loop region, the Northeast Chinese haplotypes were hardly separated from South Korean haplotypes (Fig. 3).

The TMRCA of all examined *R. ferrumequinum* individuals could be dated to 1.87 Ma (95% CI [1.39–2.35] Ma). The TMRCA estimates obtained for Clade CE China/Northeast Asia were 0.87 Ma (95% CI [0.60–1.16] Ma). For East Asian *R. ferrumequinum*, the TMRCA was 0.22 Ma (95% CI [0.12–0.34] Ma), which could be traced back to the late Pleistocene.

## Historical demography

Mismatch distribution analysis based on cyt *b* revealed different historical demography. Japanese and South Korean populations failed to reject the model of population expansion based on Hri, SSD ($P_{SSD} > 0.05$, $P_{Hri} > 0.05$) (Table 3) and their smooth or unimodal mismatch distributions (Fig. S1). The significant negative Fu's *Fs* value also indicate a sudden expansion. The most recent expansion times were estimated to be 0.15 Ma (95% CI [0.05–0.27] Ma) and 0.12 Ma (95% CI [0.03–0.20] Ma) for Japanese and South Korean populations, respectively. However, the high Hri, non-significant positive Fu's *Fs* value and multiple mismatch distribution of Northeast Chinese population suggests a stable population history or population bottlenecks (*Liao et al., 2010*) (Table 3, Fig. S1).

## DISCUSSION

*Rhinolophus ferrumequinum* from Northeast Asia diverged from other populations during 0.87–0.22 Ma, corresponding to the Quaternary Pleistocene (*Ehlers, Gibbard & Hughes, 2011*). Climate fluctuations of this epoch played important roles in shaping the geographical distribution, historical demography and genetic diversification of many organisms in the Palaearctic region (*Qu et al., 2009*).
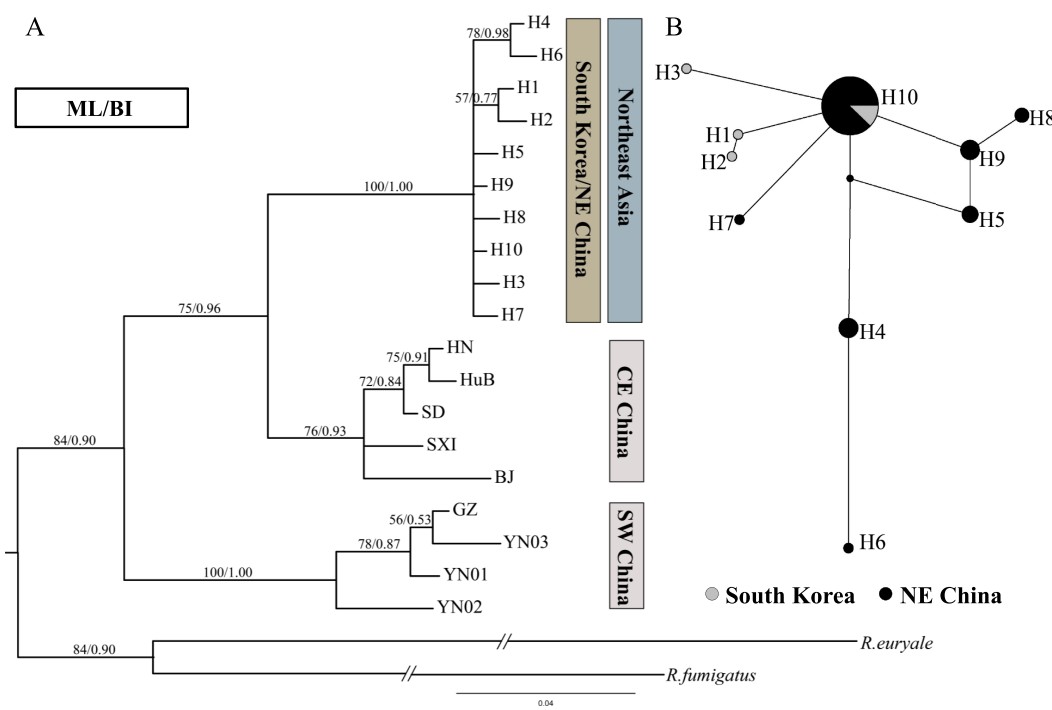

**Figure 3** Phylogenetic trees and network for *Rhinolophus ferrumequinum* populations based on D-loop haplotypes. (A) Phylogenetic trees constructed by ML and BI methods. The D-loop data of Japanese population was unavailable. (B) Median-joining network for the South Korean and Northeast Chinese *R. ferrumequinum* haplotypes. The circle size is proportional to the frequency of that haplotype. Small black dots represent missing haplotypes. Locality codes and haplotype are described in Tables S1 and S2, respectively.

**Table 3** Results of mismatch distribution analyses and neutrality tests for *Rhinolophus ferrumequinum* based on cyt *b* sequences.

|  | SSD | Hri | Tau (95%CI) | *t* (95%CI) | Tajima's *D* | Fu's *Fs* |
|---|---|---|---|---|---|---|
| NE China | 0.04 | 0.74 | 3.00 (0.55–3.00) | - | −0.84 | 0.81 |
| South Korea | 0.07 | 0.35 | 1.71 (0.43–2.98) | 0.12 Ma (0.03–0.20 Ma) | −1.73[*] | −5.26[**] |
| Japan | 0.04 | 0.16 | 2.27 (0.75–3.94) | 0.15 Ma (0.05–0.27 Ma) | −0.73 | −5.58[**] |

**Notes.**

Hri, raggedness index; SSD, sum of squared deviations; NA, data deficiencies; -, no expansion was detected.

[*]Statistically significant results are indicated by $P < 0.05$.

[**]Statistically significant results are indicated by $P < 0.01$.

In this study, the TMRCA of the greater horseshoe bats from Northeast Asia could date back to 0.22 Ma (95% CI [0.12–0.34] Ma), during the Saale glaciation (0.13–0.30 Ma) (*Lisiecki & Raymo, 2005*). Accompanied with temperature dropping in this period, the sea level declined gradually, and reached the lowest (about 130 m lower than it is today) at 0.14 Ma (*Rohling et al., 1998*; *Molodkov & Bolikhovskaya, 2002*), which could be beneficial for bats to cross the sea. *Flanders et al. (2011)* suggested that greater horseshoe bats are most likely to have originated in the Shandong Province of China. Therefore, we inferred that *R. ferrumequinum* might disperse to Northeast China and South Korea

from Shandong Province by the Bohai Sea, as this is the shortest way and the Bohai Sea would have disappeared if the sea level dropped 120 m (*Ray & Adams, 2001*). Furthermore, haplotype network seemed to support this scenario, where H15 is a more likely ancestor haplotype occupying the center of the network with numerous connects (Fig. 2B).

In Northeast China, the population might have undergone a founder effect due to its low genetic diversity. However, a specific haplotype H14 was detected in Northeast China, which may be not congruent with this notion. Instead, combining the high Hri and non-significant positive Fu's $Fs$ value, we inferred a bottlenecked event may have occurred in Northeast China (*Liao et al., 2010*). In regard to the South Korean population, we compared previous studies on *R. ferrumequinum* (*Flanders et al., 2011*; *Sun et al., 2013*; *Koh et al., 2014*) and found an expanding event was first detected at 0.12 Ma (95% CI [0.03–0.20] Ma), which corresponds to the relatively warm Eemian interglaciation (0.12–0.13 Ma) in the Pleistocene

The Changbai Mountains are the boundary of Northeast China and Korean Peninsula. Extremely low genetic divergence and high gene flow level between populations from Northeast China and South Korea suggest that the Changbai Mountains have not acted as a geographic barrier. In previous studies, *Sun et al. (2013)* and *Flanders et al. (2011)* found the Qinling Mountains have played an important role in forming different lineages of *R. ferrumequinum* bats; however, *Rossiter et al. (2007)* considered the Pyrenees (above 2,000 m) have not hindered gene flow of *R. ferrumequinum* and *Bilgin et al. (2009)* showed the Taurus Mountains and eastern Anatolian Diagonal Mountain Chain have not limited the western clade of *R. ferrumequinum* bats' distribution. Therefore, the isolation effect of different mountains is variable. Moreover, we cannot rule out other reasons, such as incomplete lineage sorting and ancestral polymorphism, which can also cause low divergence between populations from Northeast China and South Korea (*Flanders et al., 2009*).

In our study, the Japanese population formed a single sub-clade and diverged more recently than the populations from Northeast China and South Korea (Fig. 2A), which is in contrast to the *Flanders et al. (2009)* and *Flanders et al. (2011)* studies. *Flanders et al. (2009)* considered that *R. ferrumequinum* colonized East China from Japan. However, our results indicate that the Japanese population colonized more recently from Eurasian continent. It was determined that the Korea Strait is about 130 m deep, so the land bridge can only be formed during main glacial period (*McKay, 2012*). *Ohshima (1990)* mentioned that the Korean Peninsula-Japanese land bridge was estimated to have remained in place until 0.15 Ma (also see *Watanobe, Ishiguro & Nakano, 2003*). Thus, we inferred that the emergence of the land bridge favored some *R. ferrumequinum* bats to colonize Japan from Northeast China and South Korea. Additionally, other mammals, such as the Japanese wild boar (*Sus scrofa leucomystax*) (*Watanobe, Ishiguro & Nakano, 2003*), sika deer (*Cervus nippon*) (*Nagata et al., 1999*) and Asian black bear (*Ursus thibetanus*) (*Ohnishi et al., 2009*) were found to colonize Japan from Eurasian continent via the Korean Peninsula–Japanese land bridge (*Flanders et al., 2011*).

However, some studies showed that Japan was not connected to Eurasian continent during this period (*Park et al., 2000*; *Ray & Adams, 2001*; *Flanders et al., 2009*). There was

a narrow seaway (about 20 km wide) in Korea Strait connecting the East China Sea and the East Japan Sea (*Park et al., 2000*). Bats of *R. ferrumequinum* are able to fly up to 30 km between the winter and summer roosts, with the longest recorded movement being 180 km (*Paz, Fernandez & Benzal, 1986*). *Bilgin et al. (2009)* found the Marmara Sea (70 km) does not seem to limit the dispersal in *R. ferrumequinum*. Although a narrow seaway would have been present, *R. ferrumequinum* from the Eurasian continent could go through the Korea Strait and enter Japan during the glacial period. *Rhinolophus ferrumequinum* in Japan might have expanded at 0.15 Ma (95% CI [0.05–0.27] Ma), which is consistent with the expansion time (0.13–0.19 Ma) calculated by *Flanders et al. (2011)*. This expansion time is in the Saale glaciation, which suggests Japan might act as a refuge for mammals in Northeast Asia during glacial periods.

With the arrival of the Eemian interglaciation (0.12–0.13 Ma), the temperature increased gradually, resulting in rising sea levels. The Korea Strait became a natural barrier which isolated Japanese *R. ferrumequinum* from other Eurasian continental populations. Other studies also show gene flow can be hindered by water bodies, such as the Taiwan Strait (131 km) and English Channel (100 km) (*Chen et al., 2006*; *Rossiter et al., 2007*). The wider Korea Strait (180 km) was inferred to play an important role in acting as a barrier to hinder the gene flow between Japanese and Eurasian continental populations.

## ACKNOWLEDGEMENTS

We would like to thank Tinglei Jiang and Guanjun Lu who worked hard with us in the field to collect the samples used in this study. We are especially grateful to Katy Parise for her kind help with language modification. We would also like to thank Ying Tang for the lab work.

### Funding

This work was supported by the National Natural Science Foundation of China (Grant Nos. 31370399, 30900132 and 31270414), Specialized Research Fund for the Doctoral Program of Higher Education (Grant No. 20120043130002) and the Fundamental Research Funds for the Central Universities (Grant No. 2412016KJ045). The funders had no role in study design, data collection and analysis, decision to publish, or preparation of the manuscript.

### Grant Disclosures

The following grant information was disclosed by the authors:
The National Natural Science Foundation of China: 31370399, 30900132, 31270414.
Specialized Research Fund for the Doctoral Program of Higher Education: 20120043130002.
The Fundamental Research Funds for the Central Universities: 2412016KJ045.

### Competing Interests

The authors declare there are no competing interests.

## Author Contributions

- Tong Liu performed the experiments, analyzed the data, wrote the paper, prepared figures and/or tables.
- Keping Sun conceived and designed the experiments, contributed reagents/materials/-analysis tools, wrote the paper, reviewed drafts of the paper.
- Yung Chul Park and Jiang Feng reviewed drafts of the paper.

## Animal Ethics

The following information was supplied relating to ethical approvals (i.e., approving body and any reference numbers):

National Animal Research Authority in Northeast Normal University, China (approval number: NENU-20080416).

## DNA Deposition

The following information was supplied regarding the deposition of DNA sequences:

GenBank nos. KX237527–KX237538 and KX237546.

## Data Availability

The raw data has been supplied as Supplemental Information.

## Supplemental Information

Supplemental information for this article can be found online at http://dx.doi.org/10.7717/peerj.2472#supplemental-information.

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
