# Peer review of "Phylogenetic relationships and evolutionary history of the greater horseshoe bat, Rhinolophus ferrumequinum, in Northeast Asia"

_PeerJ, doi:10.7717/peerj.2472_

## Round 0.1 · original submission · Minor Revisions

We have received three reviews, and two of the reviewers felt that your paper, once suitably revised, could make a useful contribution to the literature on the evolutionary history and phylogeography of this taxon. The reviewers provide some constructive advice that will help guide your revisions. I would emphasize that two of the reviewers questioned your choice of outgroup and also felt that the comparisons and discussion relative to other works on the same species could be improved. Another point raised by two reviewers is that it would be a more valuable work if it included nuclear genetic data (such as microsatellites). I agree with this comment, and as the field of molecular ecology matures, it will be increasingly difficult to publish papers based on the inference from a single marker.

Reviewer 1 ·

Basic reporting

The article did not include sufficient introduction and background to demonstrate how the work fits into the broader field of knowledge.

And no gene sampling information mentioned in the introduction.

Fig. 3 was even not mentioned in the text.

Experimental design

The research question in this MS is not meaningful. No much contribution for the knowledge gap. Two of the three aims have been studied previously although very few samples were used before.

In 2016! the current dataset, comprising only mtDNA sequences which often lead to discordant patterns with ncDNA dataset, does not constitute a solid ground for a scientific study. Suggest to add dataset from some nuclear markers (e.g. intron or microsatellite) and combine datasets from all published papers (Rossiter et al. 2000; Rossiter et al. 2007; Flanders et al. 2009; Flanders et al. 2011; Sun et al. 2013).

No any innovation in the mythology! And no details about BEAST analysis were provided.

Samples used in the MS are not consistent for cytb and D-loop and no D-loop sequences for samples from Japan were generated.

Validity of the findings

This MS is just specific to a species (from a very restricted region) and no any general patterns were concluded.

In "Historical demography", why not show the mismatch distribution patterns (one or several peaks) for those three populations. Based on the nucleotide diversity of these three populations, i doubt the current mismatch distribution results.

Additional comments

Materials and methods
"A total of 85 sequences based on cyt b gene (1140 bp) and 62 sequences based on D-loop gene (465 bp) were obtained and analyzed (Table S1, supporting information).". This should be put into the Results section.

In phylogenetic analysis, why not use R. luctus as the outgroup; in several studies R. ferrumequinum was considered as an older species than R. pusillus and R. pearsoni that were currently used as outgroups in the MS.

Results
The authors did not explain why no D-loop sequences were obtained for samples from Japan.
The authors only included TMRCA for East Asia, but did not mention TMRCA for other clades although they were all shown in the Figure.

Reviewer 2 ·

Basic reporting

Coordinates of the sites should be provided in Table S1.

Experimental design

For the phylogenetic reconstruction, Rhinolophus pearsoni and R. pusillus have been used as outgroups but these 2 species belong to the Asian clade of Rhinolophus and hence are quite divergent from R. ferrumequinum belonging to the Afro-Paleartic clade. Have species quite distant might hinder alignment (especially for the D-loop region) and complicate phylogenetic reconstruction. It would be best to use as an outgroup a few species from the Afro-Paleartic clade, for example, species from the fumigatus, capensis or euryale groups (see Dool et al. 2016).

Dool SE, Puechmaille SJ, Foley NM, Allegrini B, Bastian A, Mutumi GL, Maluleke TG, Odendaal LJ, Teeling EC, Jacobs DS (2016) Nuclear introns outperform mitochondrial DNA in intra-specific phylogenetic reconstruction: lessons from horseshoe bats (Rhinolophidae: Chiroptera). Molecular Phylogenetics and Evolution, 97, 196-212.


Lines 164-165 and Table 2. It is unclear how these gene flow measures have been carried out as such are not mentioned in the M&M. Please, specify.

Validity of the findings

A very relevant paper that could be added in the discussion in regards to barriers to gene flow is Bilgin et al (2009) in which the impact of the Marmara sea was investigated.

Bilgin R, Çoroman E, Karataş A, Morales JC (2009) Phylogeography of the greater horseshoe bat, Rhinolophus ferrumequinum (Chiroptera: Rhinolophidae), in southeastern Europe and Anatolia, with a specific focus on whether the Sea of Marmara is a barrier to gene flow. Acta Chiropterologica, 11, 53-60.

Additional comments

This is a nice piece of work that deserves publication.

Reviewer 3 ·

Basic reporting

This is an interesting manuscript trying to resolve the demographic history of Rhinolophus ferrumequinum in Northeast China, South Korea and Japan. Overall the quality of the English is good and nothing that can't be corrected by the editorial team. One slight problem I have with the paper is its use of "East Asia". Technically East Asia covers a large geographic area (including all of China) - the present study focusses on North East China, South Korea (no samples were taken from North Korea) and Japan. I would suggest that the authors define more clearly their sampling area throughout the manuscript as it is inaccurate to say "little is known about this species from East Asia".

Minor points:

Line 43 onwards - I think it is important that you clarify what region of the mitochondria the different authors (Flanders et al and Sun et al) are using because these studies look at ND2 and cyt-b respectively which may have slight implications (e.g. in mutation rate) when analysing the results).

Line 60 - This is the first time you mention Koh et al. (2014) so it would be useful to the reader that you clarify that this study is on R. ferrumequinum, and what region of the mitochondria they used in their study.

Line 72-74 - For this section it would be better if you just referred to the samples you collected and not the ones you downloaded off genbank as the DNA extraction and amplification may be different - better that you talk about your own samples and say in the results how many samples you included from previous studies.

Line 114-118 - These two sentences need to be switched around (move the second sentence to the start of the paragraph) as it is unclear what NETWORK is at the start of the paragraph.

Line 123 - can you be more specific what species were used in these studies rather than just reporting the Genus.

Line 192 - I am unsure what you mean by the bats "occurred in 0.07-1.23 Ma", please rephrase.

Experimental design

The experimental designed followed the standard tests to determine population genetic structure and demographic history. It is disappointing that the authors were not able to include microsatellite analyses in this study as they would have been able to include what Sun et al (2013) had already genotyped but this is a minor point. It is also a shame that no Japanese samples were sequenced at the d-loop.

Validity of the findings

The data is robust and statistically sound. However, I feel that the authors could have used the data they have produced more effectively. Considering there have been papers before this looking at the phylogrography of R. ferrumequinum I feel that more could have been made comparing the results of the different papers and speculating as to why there are differences between them (e.g. TMRCA). The authors do highlight the similarities between the timing of population expansion of the Japanese population and how this was found elsewhere but considering they did not detect any Japanese haplotypes on mainland China (as reported by Flanders et al 2011) it would have been interesting to compare the different studies.

---

## Round 0.2 · Minor Revisions

I am generally satisfied with authors responses to the reviewer comments and the changes made to the manuscript. However, I do think the paper could benefit from some more careful proof-reading and attention to English usage. If possible, please have a native English speaking colleague go over the manuscript one more time to make sure it is clean and grammatically correct prior to resubmission and eventual publication.

---

## Round 0.3 · accepted · Accept

Thank you for providing an improved edited version of your manuscript. I am pleased to say that it is now suitable for publication in PeerJ.